# Postoperative Urinary Retention after Pediatric Orthopedic Surgery

**DOI:** 10.3390/children9101488

**Published:** 2022-09-28

**Authors:** Mohan V. Belthur, Ian M. Singleton, Jessica D. Burns, M’hamed H. Temkit, Thomas J. Sitzman

**Affiliations:** 1Department of Orthopedics, Phoenix Children’s Hospital, Phoenix, AZ 94304, USA; 2College of Medicine, University of Arizona—Phoenix, Phoenix, AZ 85004, USA; 3Department of Surgery, Mayo Clinic College of Medicine, Scottsdale, AZ 85259, USA; 4Department of Clinical Research, Phoenix Children’s Hospital, Phoenix, AZ 94304, USA; 5Division of Plastic Surgery, Phoenix Children’s Hospital, Phoenix, AZ 94304, USA

**Keywords:** cerebral palsy, urinary retention, postoperative, pediatric, orthopedic surgery

## Abstract

Purpose: This study aims to describe the incidence of postoperative urinary retention among pediatric patients undergoing orthopedic surgery and identify risk factors. Methods: The Pediatric Health Information System was used to identify children aged 1–18 years who underwent orthopedic surgery. Collected from each patient’s record were demographic information, principal procedure during hospitalization, the presence of neurologic/neuromuscular conditions and other complex chronic medical conditions, the total postoperative length of stay, and the presence of postoperative urinary retention. Results: The overall incidence of postoperative urinary retention was 0.38%. Children with complex chronic neuromuscular conditions (OR 11.54 (95% CI 9.60–13.88), *p* = < 0.001) and complex chronic non-neuromuscular medical conditions (OR 5.07 (95% CI 4.11–6.25), *p* ≤ 0.001) had a substantially increased incidence of urinary retention. Surgeries on the spine (OR 3.98 (95% CI 3.28–4.82, *p* ≤ 0.001) and femur/hip (OR 3.63 (95% CI 3.03–4.36), *p* ≤ 0.001) were also associated with an increased incidence. Conclusions: Children with complex chronic neuromuscular conditions have a substantially increased risk of experiencing postoperative urinary retention. Complex chronic non-neuromuscular medical conditions and surgeries to the spine, hip, and femur also carry a notably increased risk.

## 1. Introduction

Postoperative urinary retention (POUR) is defined as an inability to void after surgery or the presence of a major residual volume after voiding that requires catheterization [1,2,3,4]. POUR causes significant patient discomfort and usually requires treatment with bladder catheterization, which is associated with prolonged hospital stay, risk of urinary tract infection, and ultimately increased healthcare costs. Untreated POUR can lead to severe complications, including sepsis [1,2,3,5,6,7].

For adult patients undergoing orthopedic surgery, the rate of POUR varies between 9–84% depending on the specific surgery and the exact definition of POUR used [1,8,9]. Risk factors have been studied extensively and include increasing age, a history of anticoagulant medication, epidural analgesia, use of patient-controlled analgesia, and a history of diabetes mellitus [1,8,10,11,12,13,14,15,16].

In contrast to adults, the overall incidence of POUR in pediatric orthopedic surgery has not been well-characterized. Single-center studies have reported the incidence of urinary retention at 29% among children undergoing lower extremity surgery and 46% among young patients who underwent posterior spinal fusion [17,18]. In addition to being limited by small samples, these studies provide no information on the overall incidence of urinary retention after pediatric orthopedic surgery or how incidence rates vary among different procedures.

Research is also lacking on the impact of pre-existing medical conditions on rates of urinary retention after pediatric orthopedic surgery. Brenn et al. [19] found a 70% incidence of POUR among pediatric cerebral palsy patients after use of epidural analgesia, and it seems likely to be a risk factor for urinary retention after orthopedic procedures. The presence of chronic medical conditions has also been documented in the literature as associated with POUR in adults, particularly renal failure and diabetes mellitus with diabetic complications [5]. While the same relationships may exist for pediatric patients, the influence of these medical conditions and other patient-specific factors on the risk for urinary retention is poorly understood in the pediatric orthopedic surgery population. In addition, although POUR is associated with an increased length of stay in adults, the impact on pediatric orthopedic patients is less well characterized [5]. This lack of knowledge prevents targeted efforts at avoidance and early detection.

The present study evaluates the incidence of POUR among children undergoing orthopedic surgery at 49 free-standing children’s hospitals participating in the Pediatric Health Information System (PHIS). These hospitals provide 85% of all pediatric specialty care delivered in the United States [20]. The study has three distinct objectives: (1) to determine the incidence of POUR among children undergoing orthopedic surgery, (2) to identify surgical procedures and patient risk factors associated with an increased risk of POUR, and (3) to evaluate the impact of POUR on inpatient hospital stay following orthopedic surgery.

## 2. Materials and Methods

PHIS was used to perform a retrospective cohort study of children aged 1–18 years who underwent orthopedic surgery between 2012 and 2015. PHIS is a large administrative database maintained by the Children’s Hospital Association that includes clinical, demographic, and billing information for all hospital discharges occurring in a network of 49 free-standing children’s hospitals located in the United States. The data is based on billing charges, and all discharges for each hospital stay are included in the database [21]. The Institutional Review Board at the authors’ institution reviewed this study and determined it was not human subjects research, as defined by the Common Rule (45CFR46.102[f]) because the data set was deidentified.

Eligible children were identified using International Classification of Disease, Ninth Revision (ICD-9) procedure codes for operations on the musculoskeletal system (ICD-9 codes 77–84). For children with more than one encounter during the study period, only the first encounter was included. 

For all eligible children, information was collected on the child’s demographics, medical conditions, and surgical care. Demographic information included age at admission, gender, race, ethnicity, median household income by zip code of residence, and primary source of payment. Also collected was the principal procedure during hospitalization, all diagnosis codes for the hospitalization, the presence of surgical complication flags (e.g., non-healing surgical wound, hemorrhage during procedure, seroma formation), medical complication flags (e.g., adverse effects of anesthesia, anaphylactic shock due to other serum, postvaccination fever), infection flags (e.g., *Shigella dysenteriae*, cryptosporidiosis), neurologic/neuromuscular flags (e.g., brain and spinal cord malformation, central nervous system disease, muscular dystrophies and myopathies), complex chronic conditions flags (e.g., heart and great vessel malformation, chronic kidney disease, cystic fibrosis), cerebral palsy (ICD-9-CM code 343.xx), and the total postoperative length of stay. Flags were applied to each patient’s profile by PHIS and were not determined by the study authors [22]. The location of the surgery was determined by the principal procedure. The outcome variable was the occurrence of POUR as identified by the presence of the diagnosis code for the inability to urinate or incomplete emptying of the bladder (ICD-9 code 788.2). 

### Statistical Analysis

Preliminary group comparisons were conducted using the two independent samples t-test for continuous variables and the chi-squared test for categorical variables. Simple logistic regression was performed to identify risk factors associated with POUR. A multivariable logistic regression model was developed incorporating risk factors identified from the simple logistic regression to model the odds of POUR. The results were summarized using the odds ratio, corresponding 95% confidence interval (95% CI), and the p-value. The C-statistic, which is the area under the receiver operating characteristic curve (AUROC), was reported as a measure of model discrimination, with a higher C-statistic indicating how well the model is able to accurately predict POUR versus not. The significance level was set a priori at 0.05. Statistical analyses were performed using the statistical software package SAS 9.4 (SAS Institute, Cary, NC, USA).

## 3. Results

A total of 232,551 pediatric orthopedic surgery patients met the inclusion criteria. The mean age at admission was 9.2 years, and 55.6% of patients were male. The majority of patients were white at 62.5% of the sample; 19.6% of patients were of Hispanic-Latino ethnicity. Chronic neuromuscular conditions were present in 6.0% of patients, while 10.0% had complex chronic medical conditions that were not neuromuscular. Prevalence of relevant patient demographic and medical conditions are presented in Table 1. 

POUR occurred following 892 procedures, resulting in an overall incidence of 0.38%. Patients with urinary retention were much more likely to have a chronic neuromuscular condition, which included cerebral palsy, spina bifida, and muscular dystrophy (Table 2). Patients with cerebral palsy comprised 24.8% of all POUR cases in the present study.

The most common surgical locations that subsequently developed POUR were the spine, femur, and hip (Figure 1). The most common surgeries associated with POUR were posterior spinal fusion for spine deformity and hip reconstruction surgery.

On univariable analysis, the risk of POUR was highest among patients with a chronic neuromuscular condition. The risk was also increased by the presence of a non-neuromuscular complex chronic condition. If a surgical complication, medical complication, or infection occurred during the patient’s admission, this also increased the risk of POUR. White race was also associated with an increased risk of POUR (Table 3). Increased age at admission was also associated with a higher risk of urinary retention; the average age of patients who developed POUR was 11.9 years compared to 9.2 years in children who did not (*p* < 0.001). 

The average length of stay in patients who had urinary retention was 7.8 days compared to 1.7 days in patients who did not (Figure 2). 

On multivariable analysis, significant risk factors for POUR in order of decreasing odds ratio were presence of neuromuscular complex chronic conditions, presence of non-neuromuscular complex chronic conditions, surgery on the spine, surgery on the femur/hip, male gender, white race, and increasing age at admission. (Table 4). A risk prediction profile developed with multivariable logistic regression analysis using the above-identified risk factors returned a C-statistic of 0.87.

## 4. Discussion

For the pediatric population receiving orthopedic surgery, the presence of chronic medical conditions and the surgery location are strong predictors for developing POUR. The risk of developing POUR is particularly elevated for patients with neuromuscular conditions: presence of a neuromuscular condition was associated with an 11.5-fold higher odds of POUR. While children with cerebral palsy represented only 3.8% of the study population, they accounted for one-fourth of all patients who developed POUR. While only 0.38% of all children developed POUR, the incidence of POUR was almost 4-fold higher for children undergoing femur or hip surgery and almost 4-fold higher for children undergoing spine surgery. These findings suggest surgeons should focus their concern for POUR on these select patient populations. 

The risk factors identified in this study are similar to those documented in the literature. Sherburne et al. [17] found that older age increased the risk of POUR in pediatric patients undergoing lower extremity orthopedic surgery, while Keskinen et al. [18] found male gender to be associated with POUR in pediatric patients undergoing posterior spinal fusion for idiopathic scoliosis. The present study found these two factors to increase the risk of urinary retention for all pediatric orthopedic surgery patients. However, Sherburne et al. did not find gender to be significantly associated with POUR, which conflicts with both the results of Keskinen et al. as well as our own. Furthermore, Sherburne et al. did not find a history of neurogenic bladder or previous bladder problems to be significantly associated, while our results found an association between neuromuscular conditions, particularly cerebral palsy, and POUR. These conflicting findings may be due to differences in patient population—the present study included all pediatric orthopedic surgeries, while Sherburne et al. studied only pediatric lower extremity surgeries. 

Patients with cerebral palsy represented one-fourth of all patients with POUR in the present study, and the presence of neuromuscular conditions had the highest odds of developing POUR out of any risk factor. Urinary tract dysfunction is common in cerebral palsy patients, with Brenn et al. [19] finding a 70% incidence of POUR among pediatric cerebral palsy patients receiving epidural anesthesia [23,24,25,26]. Thus, it seems likely that cerebral palsy is indeed a significant risk factor. The presence of chronic medical conditions also has been documented in the literature as associated with POUR in adults, particularly renal failure and diabetes mellitus with diabetic complications [5]. This study found a similar relationship between medical comorbidities and POUR. 

The incidence of POUR in this study is less than that reported in single-center studies in the literature [17,18]. Patients who underwent surgeries performed on the spine and femur/hip comprised 38% and 22%, respectively, of the patients who experienced POUR in this study. However, only 2.9% of all of the patients who underwent spine surgery and 1.2% of the patients who underwent femur/hip surgery experienced POUR. Sherburne et al. [17] reported that of the 38 pediatric patients in their sample who underwent lower extremity orthopedic surgery, 29% required postoperative straight catheterization. Keskinen et al. [18] reported that of their sample of 111 young patients (age 11–21 years) who underwent posterior spinal fusion for adolescent idiopathic scoliosis, 46% required postoperative intermittent catheterization. These different findings may be due to the use of ultrasound to accurately assess bladder volume as well as frequent monitoring and reporting of postoperative bladder volume in single-center, procedure-focused studies. Spine and femur/hip procedures in pediatric patients can require long durations of anesthetic and postoperatively be painful and require increased narcotic use [27,28]. Longer surgery time is known to increase complications, including POUR, and is likely responsible for the increased rate with which patients who underwent spine and femur/hip operations in this study experienced POUR [29]. 

In regard to white race increasing the odds of developing POUR, it is likely that this is due to socioeconomic factors. White children may go to hospitals or medical centers with increased resources available. This may translate into increased monitoring of postoperative bladder volumes and subsequent diagnosing of POUR. For example, Stockwell et al. [30] found Latino children to experience almost double the rate of adverse events as compared to white children. 

Ultimately, POUR is a significant complication that substantially increases the cost of the healthcare provided. Wu et al. [5] report that adult patients who underwent total hip or knee arthroplasty and subsequently developed POUR stayed an extra 0.6 and 0.48 days in the hospital, respectively. Fernandez et al. [15] found that adult patients who developed POUR after total hip or knee arthroplasty had a mean length of stay of 6.7 days compared to 4.6 days without retention. In our study, the average length of stay of patients who had urinary retention was 7.8 days compared to 1.7 days in patients who did not develop POUR, an additional length of stay that is far greater than that found in adults with POUR. This suggests that pediatric orthopedic patients who develop POUR either take longer to resume normal micturition or that clinicians are more inclined to observe them for longer periods than adult patients, possibly due to the lack of a defined POUR protocol. With an estimated hospital-adjusted expense of USD 2653 per day of inpatient admission for non-profit hospitals in the United States, the six additional days that pediatric patients with POUR remain in the hospital likely substantially increases healthcare cost and occupies key resources and personnel [31].

At the authors’ institution, a protocol is in place to monitor and treat POUR. Typically, a Foley is maintained in place until postoperative day two. High-risk patients will be followed closely with bladder ultrasound scans to monitor any development of POUR. If necessary, a straight catheter will be used to drain the bladder. If a straight catheter insertion is required more than twice, the patient is considered to have developed POUR, and a Foley is reinserted and maintained for an additional seven days. Therefore, it is important to be aware of the risk factors for POUR in order to determine which patients are at the highest risk and require the closest monitoring. 

POUR following pediatric orthopedic procedures is a relatively uncommon complication that increases patient discomfort and length of stay in the hospital. This risk will be validated prospectively in future studies and will be used to implement a protocol for awareness, prevention, and management of POUR in the pediatric population following orthopedic surgery.

There are several limitations to the study. On 1 October 2015, the United States switched from the ICD-9 to the ICD-10 coding system. This increased the complexity of the coding system overall. For example, the ICD-9 code procedure code 77.01 has 21 equivalent ICD-10 codes. While the ICD-10 procedure codes are much more precise and a much-needed improvement over ICD-9, this exponential increase in the number of codes limited the analysis to ICD-9 codes only. Other limitations of this study are inherent to the approach. First, as in the case of any research using administrative data, this study relied on accurate ICD-9-CM coding for the identification of cases and characteristics. Local coding practice can influence the accuracy of diagnostic coding, and this bias cannot be excluded. This is particularly important for the ICD codes associated with POUR, whose inclusion in the database relies on accurate diagnosis by the treating physician and on correct coding for a relatively uncommon diagnosis. Both factors suggest that POUR is underreported in the present study. However, since factors leading to underreporting of urinary retention are consistent across all patients, the relative estimates of POUR presented in this study are likely unbiased. In addition, although on multivariate analysis, chronic medical conditions were found to be associated with POUR, the chronic medical condition flag from the PHIS database was used in order to classify the presence of chronic medical conditions. This did not allow for the determination of specific chronic medical conditions responsible for the development of POUR and is a potential area of focus for future research. 

The incidence of POUR in this study was 0.38% in children ≤ 18 years of age. The presence of complex chronic medical conditions, and in particular neuromuscular conditions including cerebral palsy, spina bifida, and muscular dystrophy, is a strong risk factor for developing POUR in pediatric orthopedic surgery patients. Procedures with the highest risk of developing POUR include femur, hip, and spine surgeries. POUR is associated with a significantly increased length of stay. As chronic damage can occur with even one episode of overdistention, clinicians would be well-served to monitor the postoperative urination and bladder volume of pediatric patients with these risk factors.

## Figures and Tables

**Figure 1 children-09-01488-f001:**
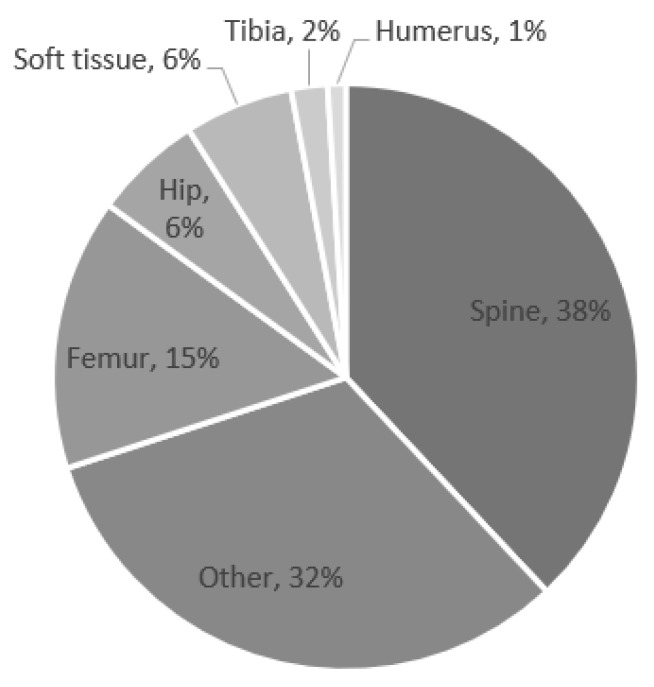
Percentage of postoperative urinary retention by location.

**Figure 2 children-09-01488-f002:**
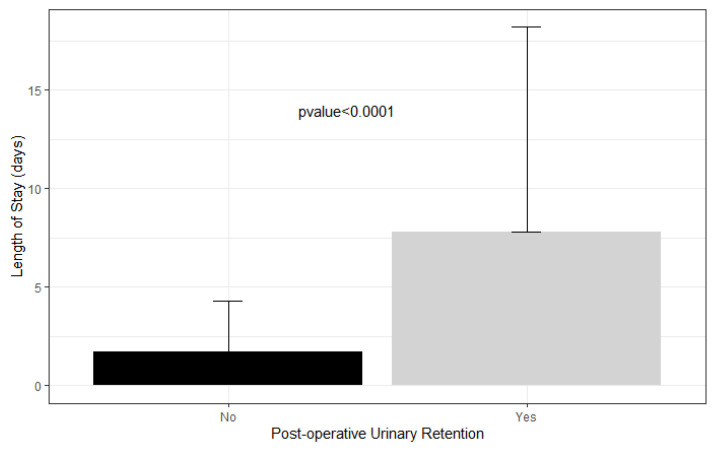
Length of stay in patients with postoperative urinary retention versus not.

**Table 1 children-09-01488-t001:** Risk factors by postoperative urinary retention.

Risk Factor	Postoperative Urinary Retention		*p* Value
	No (*n* = 231,659)	Yes (*n* = 892)	
**Male**			0.22031
No	102,891 (44.4%)	378 (42.4%)	
Yes	128,742 (55.6%)	514 (57.6%)	
**Race**			**<0.0001**
Asian	5550 (2.4%)	17 (1.9%)	
Black	33,246 (14.4%)	108 (12.1%)	
Native American	1365 (0.6%)	12 (1.3%)	
White	144,809 (62.5%)	617 (69.2%)	
Other	46,689 (20.2%)	138 (15.5%)	
**White**			**<0.0001**
No	86,850 (37.5%)	275 (30.8%)	
Yes	144,809 (62.5%)	617 (69.2%)	
**Ethnicity**			**0.0001**
Hispanic or Latino	45,307 (19.6%)	133 (14.9%)	
Not Hispanic or Latino	167,053 (72.1%)	700 (78.5%)	
Unknown	19,299 (8.3%)	59 (6.6%)	
**Age at Admission (years)**			**<0.0001**
N	23,1659	892	
Mean (SD)	9.2 (5.0)	11.9 (4.3)	
Range	(1.0–18.0)	(1.0–18.0)	
**Primary Source of Insurance**			0.2510
Government	158,404 (68.4%)	590 (66.1%)	
Private	69,325 (29.9%)	289 (32.4%)	
Other	3930 (1.7%)	13 (1.5%)	
**Median Household Income ($)**			**0.0003**
N	231,659	892	
Mean (SD)	47,766.8 (18,851.9)	49,864.4 (19,366.6)	
Range	(6320.0–196,032.0)	(6320.0–139,915.0)	
**Surgical Location**			**<0.0001**
Spine	11,601 (5.0%)	340 (38.1%)	
Femur	13,316 (5.7%)	137 (15.4%)	
Hip	3363 (1.5%)	58 (6.5%)	
Tibia	5834 (2.5%)	16 (1.8%)	
Humerus	6781 (2.9%)	9 (1.0%)	
Soft tissue	12,035 (5.2%)	50 (5.6%)	
Other	178,729 (77.2%)	282 (31.6%)	
**Medical Complication Flag**			**<0.0001**
No	231,490 (99.9%)	884 (99.1%)	
Yes	169 (0.1%)	8 (0.9%)	
**Surgical Complication Flag**			**<0.0001**
No	226,091 (97.6%)	601 (67.4%)	
Yes	5568 (2.4%)	291 (32.6%)	
**Infection Flag**			**<0.0001**
No	222,328 (96.0%)	751 (84.2%)	
Yes	9331 (4.0%)	141 (15.8%)	
**Complex Chronic Condition (CCC)**			**<0.0001**
Neuromuscular CCC	13,618 (5.9%)	333 (37.3%)	
Non-neuromuscular CCC	23,033 (9.9%)	325 (36.4%)	
No CCC	195,008 (84.2%)	234 (26.2%)	
**Cerebral Palsy**			**<0.0001**
No	223,022 (96.3%)	671 (75.2%)	
Yes	8637 (3.7%)	221 (24.8%)	
**Length of Stay (days)**			**<0.0001**
N	231,659	892	
Mean (SD)	1.7 (2.6)	7.8 (10.4)	
Range	(1.0–259.0)	(1.0–128.0)	

Bold = statistically significant; CCC = complex chronic condition.

**Table 2 children-09-01488-t002:** Percentage of patients with the neurological and neuromuscular flag who had cerebral palsy, spina bifida, or muscular dystrophy.

Risk Factor	Neurological and Muscular		Total
	Yes	No	
**Cerebral Palsy**			
Yes	66.6%	0.0%	24.8%
No	33.4%	100.0%	75.2%
**Spina Bifida**			
Yes	5.5%	0.2%	2.1%
No	94.6%	99.8%	97.9%
**Muscular Dystrophy**			
Yes	7.2%	0.0%	2.7%
No	92.8%	100.0%	97.3%

**Table 3 children-09-01488-t003:** Simple Logistic Analysis for Postoperative Urinary Retention.

Risk Factor	OR (95% CI)	*p*-Value
**Male vs. Female**	**1.09 (0.95–1.24)**	**0.2204**
**Race**		
Asian vs. White	0.72 (0.44–1.16)	0.1801
Black vs. White	0.76 (0.62–0.94)	**0.0094**
Native American vs. White	2.06 (1.16–3.66)	**0.0134**
Other vs. White	0.69 (0.58–0.83)	**0.0001**
**White vs. Not**	1.35 (1.17–1.55)	**<0.0001**
**Age at Admission (years)**	1.12 (1.11–1.14)	**<0.0001**
**Primary Insurance**		
Government vs. Other	1.13 (0.65–1.95)	0.6728
Private vs. Other	1.26 (0.72–2.2)	0.4155
**Cerebral Palsy vs. Not**	8.5 (7.29–9.91)	**<0.0001**
**Surgical Complication Flag vs. Not**	19.66 (17.05–22.67)	**<0.0001**
**Medical Complication Flag vs. Not**	12.4 (6.08–25.27)	**<0.0001**
**Infection Flag vs. Not**	4.47 (3.73–5.36)	**<0.0001**
**Complex Chronic Condition (CCC)**		
Neuromuscular CCC vs. No CCC	20.38 (17.23–24.11)	**<0.0001**
Non-Neuromuscular CCC vs. No CCC	11.76 (9.93–13.92)	**<0.0001**

Bold = statistically significant.

**Table 4 children-09-01488-t004:** Multivariable Logistic Model for Postoperative Urinary Retention.

Risk Factor	OR (95% CI)	*p* Value
Male vs. Female	1.46 (1.27–1.68)	<0.0001
White vs. Not	1.26 (1.09–1.45)	0.0018
Neuromuscular CCC vs. NO CCC	11.54 (9.6–13.88)	<0.0001
Non-Neuromuscular CCC vs. NO CCC	5.07 (4.11–6.25)	<0.0001
Spine vs. other location	3.98 (3.28–4.82)	<0.0001
Femur/Hip vs. other location	3.63 (3.03–4.36)	<0.0001
Admit Age (years)	1.06 (1.04–1.08)	<0.0001

Note: AUROC = 0.87.

## Data Availability

Data available at https://www.childrenshospitals.org/phis.

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
