# Peer review of "Postoperative Urinary Retention after Pediatric Orthopedic Surgery"

_children, 2022, doi:10.3390/children9101488_

Round 1

Reviewer 1 Report

This is an excellent observation with a large number of patients across multiple centers which adds strength. The cost analysis is helpful and could expand for added benefit, but it is overall very nice study and of interest to many aspects of our care, treatment and business of medicine.

Author Response

Thank you for your kind feedback. We greatly appreciate the enthusiasm about manuscript.

Reviewer 2 Report

I had the pleasure of reading an article on a typical pediatric postoperative issue. The current state of the art indicates that there aren't many papers that highlight the specific procedures and accompanying diseases that patients had in the past. This is a novel addition to the literature, and I suggest making the following minor changes:

- in the introduction, authors seem to begin every sentence with "postoperative urinary retention". This slightly disrupts the reading flow;

Row 38 - can authors introduce a phrase on what the current definition of urinary retention is?

- there are plenty of mentions of "urinary retention". I suggest adding an abbreviation throughout the entire manuscript text (e.g., UR);

- I suggest adding a Figure 1 that has more quality and not a pixelated one;

- more than 40% of references seem to be older than 10 years; I suggest improving them when possible.

Author Response

- in the introduction, authors seem to begin every sentence with "postoperative urinary retention". This slightly disrupts the reading flow;

Row 38 - can authors introduce a phrase on what the current definition of urinary retention is?

- there are plenty of mentions of "urinary retention". I suggest adding an abbreviation throughout the entire manuscript text (e.g., UR);

Thank you for your kind feedback. We agree with both points and have abbreviated postoperative urinary retention throughout the manuscript. The definition of postoperative urinary retention varies between institutions, but we have added a section on our institution’s definition and protocol.

- I suggest adding a Figure 1 that has more quality and not a pixelated one;

We have uploaded a new, higher quality figure.

- more than 40% of references seem to be older than 10 years; I suggest improving them when possible.

We agree; unfortunately, research on pediatric postoperative urinary retention is lacking. This is partly why we felt this paper was necessary, and why some of the references are older than would be preferred.

Reviewer 3 Report

It is a good study conducted by all the authors. The topic is relevant to the clinical practice. Most of the children who undergo spine surgery or hip surgery, they will have urinary catheterisation inserted in the operating theatre just prior to the surgery. This is due to anticipated long operating hours for the surgery. Probably the authors could discuss in this particular group of patients, how and when could we determine that they will have urinary retention since removal of the urinary catheter varies with different surgeons/place of practice. Is there any recommendation from previous literature on which day of the postoperatively day would be the best to remove the urinary catheter in order to assess the risk of urinary retention ? If there is no previous literature, what are the opinion/thought or practice of the current authors ?

The other point is that, what is the longest duration of urinary retention has ever been reported in the literature or in this current study ? Is there any investigation that could be performed such as MRI if the urinary retention persists much longer than we anticipate ?

Author Response

Thank you for your kind feedback. In our experience postoperative urinary retention can potentially have a significant duration, especially in patients with spina bifida. It would not be unexpected to trial multiple periods of Foley removal. There are case reports in the literature for patients requiring continuous catheterization postoperatively. We have added a paragraph on our institutions protocol for monitoring postoperative urinary retention. Typically, patients maintain their Foley in place until postoperative day two. High risk patients will be followed closely with bladder ultrasound scans to monitor any development of POUR. If necessary, a straight catheter will be used to drain the bladder. If a straight catheter insertion is required more than twice the patient is considered to have developed POUR and a Foley is reinserted and maintained for an additional seven days. Therefore, it is important be aware of the risk factors for POUR in order to determine which patients are at the highest risk and require the closest monitoring.